# A Self-Powered High-Responsivity, Fast-Response-Speed Solar-Blind Ultraviolet Photodetector Based on CuO/β-Ga_2_O_3_ Heterojunction with Built-In Potential Control

**DOI:** 10.3390/nano13050954

**Published:** 2023-03-06

**Authors:** Sangbin Park, Younghwa Yoon, Hyungmin Kim, Taejun Park, Kyunghwan Kim, Jeongsoo Hong

**Affiliations:** Department of Electrical Engineering, College of IT Convergence, Gachon University, 1342, Seongnam-daero, Sujeong-gu, Seongnam-si 13120, Gyeonggi-do, Republic of Korea

**Keywords:** solar-blind ultraviolet, photodetector, self-powered, heterojunction, built-in potential

## Abstract

Controlling built-in potential can enhance the photoresponse performance of self-powered photodetectors. Among the methods for controlling the built-in potential of self-powered devices, postannealing is simpler, more efficient, and less expensive than ion doping and alternative material research. In this study, a CuO film was deposited on a β-Ga_2_O_3_ epitaxial layer via reactive sputtering with an FTS system, and a self-powered solar-blind photodetector was fabricated through a CuO/β-Ga_2_O_3_ heterojunction and postannealed at different temperatures. The postannealing process reduced the defects and dislocations at the interface between each layer and affected the electrical and structural properties of the CuO film. After postannealing at 300 °C, the carrier concentration of the CuO film increased from 4.24 × 10^18^ to 1.36 × 10^20^ cm^−3^, bringing the Fermi level toward the valence band of the CuO film and increasing the built-in potential of the CuO/β-Ga_2_O_3_ heterojunction. Thus, the photogenerated carriers were rapidly separated, increasing the sensitivity and response speed of the photodetector. The as-fabricated photodetector with 300 °C postannealing exhibited a photo-to-dark current ratio of 1.07 × 10^3^; responsivity and detectivity of 30.3 mA/W and 1.10 × 10^12^ Jones, respectively; and fast rise and decay times of 12 ms and 14 ms, respectively. After three months of storage in an open-air space, the photocurrent density of the photodetector was maintained, indicating good stability with aging. These results suggest that the photocharacteristics of CuO/β-Ga_2_O_3_ heterojunction self-powered solar-blind photodetectors can be improved through built-in potential control using a postannealing process.

## 1. Introduction

Self-powered photodetectors have been extensively studied, owing to the advantage of not requiring an external power source, which is typically a kind of battery. In terms of energy, self-power supports conservation, as discarded batteries can pollute the environment and require additional cost to recycle [1]. Future self-powered photodetectors are expected to exhibit high performance, including fast response speed, high responsivity, high on–off ratio, excellent spectrum selectivity, and stability [2]. To improve the photoresponse characteristics of photodetectors, studies have been conducted on their materials, processes, and structures [3,4,5,6,7,8,9,10,11]. The properties of photodetectors can be easily adjusted with alternate materials or by developing novel materials. However, some materials are expensive or are available in limited supply, and the study of novel materials requires considerable time, effort, and cost. Thus, we adjusted the built-in potential using a postannealing process to increase the responsivity and response speed of a self-powered heterojunction-based photodetector. The built-in potential of the self-powered photodetector significantly affected the response speed and responsivity [12].

Of the wavelengths that can be detected by photodetectors, UVC (200–280 nm) has an extremely low background noise because UVC radiation generated by the sun is mainly absorbed by the ozone layer before reaching earth, known as solar-blind UV [13]. As solar-blind UV light offers low noise and interference, solar-blind photodetectors have been used in applications that require highly accurate detection, such as fire prevention, missile detection, and satellite communication [14,15].

Gallium oxide (Ga_2_O_3_) has different phases of α, β, γ, δ, and ε. Among them, β-Ga_2_O_3_ is relatively inexpensive and has an ultrawide bandgap (4.8–4.9 eV without alloying or doping) and is easily fabricated as bulk single crystals or high-quality large-area substrates compared to other ultrawide-bandgap materials [16]. With wide-bandgap energy, β-Ga_2_O_3_ is often used in UVC photodetectors [3,4,5,6,7,8,9]. In addition, β-Ga_2_O_3_ has excellent thermal and chemical stability. However, β-Ga_2_O_3_ is difficult to apply in homojunction devices, owing to difficulty in fabricating p-type β-Ga_2_O_3_. There is still a need for further research to achieve p-type Ga_2_O_3_, which makes p-n homojunction-based devices [17]. Thus, β-Ga_2_O_3_-based photovoltaic devices have been studied in the form of heterojunctions with other materials [6,7,8,9,10]. As heterojunction materials, cuprous oxide (Cu_2_O) and cupric oxide (CuO) have been extensively studied as p-type materials for gas sensing, diode applications, and optoelectronics [18,19,20,21]. As Cu_2_O is difficult to produce without a CuO phase, it is difficult to use in devices. Unlike Cu_2_O, single-phase CuO films are easily fabricated using sputtering systems. Additionally, CuO is an environmentally friendly, relatively abundant, inexpensive production material, and the electrical properties of CuO film can easily be controlled by doping other materials, such as Al-doped CuO or nitrogen-doped CuO film [22,23,24,25].

In this study, self-powered photodetectors based on p-CuO/n-β-Ga_2_O_3_ heterojunctions were produced and were postannealed via rapid thermal annealing (RTA) at different temperatures. The value of the built-in potential of the photodetectors was changed according to the annealing temperature. The CuO film annealed at 300 °C had the highest carrier concentration, producing a higher built-in potential than those of other samples [26]. A self-powered photodetector with a high built-in potential indicates a faster response speed and higher responsivity than lower built-in potential devices because the photogenerated carriers are rapidly and effectively separated by the built-in electric potential barrier [12]. An as-fabricated self-powered photodetector based on p-CuO/n-β-Ga_2_O_3_ and annealed at 300 °C produced a responsivity of 30.3 mA/W and rise and decay times of 12 ms and 14 ms, respectively, with 254-nm solar-blind UV light and zero bias. This study suggests that adjusting the built-in potential of self-powered photodetectors using a post-treatment process can improve the photocharacteristics. Other processes that can change the built-in potential would be used to increase the responsivity and response speed of self-powered photodetectors.

## 2. Experimental Section

### 2.1. Materials

The p-CuO/n-β-Ga_2_O_3_ heterojunction photodetectors were based on an n-type single crystal, a Sn-doped β-Ga_2_O_3_ wafer (with N_d_–N_a_ = 6.3 × 10^18^ cm^−3^ and a thickness of 641 μm), and an epitaxial layer of Si-doped β-Ga_2_O_3_ (with N_d_–N_a_ = 2.8 × 10^16^ cm^−3^ and a thickness of 8.6 μm) deposited via halide vapor-phase epitaxy (HVPE). Electron-beam evaporation was used for depositing Ti/Au electrodes (10/40 nm) on the backside of the Sn-doped β-Ga_2_O_3_ wafer. Epitaxial layers of β-Ga_2_O_3_ with Ti/Au electrodes were obtained from Novel Crystal Technology (Saitama, Japan). The 4-inch Ag (99.99%) and Cu (99.99%) targets for DC magnetron sputtering were acquired from RND Korea (Gwangmyeong, Republic of Korea). The soda–lime glass substrates (76 × 26 × 1 mm^3^ of microsphere glass) were purchased from Marienfeld (Lauda-Königshofen, Germany).

### 2.2. Deposition and Characterization of Ag/CuO Layer

The surface morphology, crystal structure, and electrical and optical properties of the Ag electrodes, and p-type layer of CuO, deposited on the glass substrate by the FTS system, were evaluated. The structural and optical properties were measured on the glass substrate, because sliced Ga_2_O_3_ wafers are not large enough to measure the structural properties and the ohmic electrode is deposited on the wafer, making it difficult to investigate the transmittance. For the deposition process, 4-inch Ag and Cu targets were used. The base pressure was 3 × 10^−5^ Torr and the working pressures were set to 2 mTorr and 2.5 mTorr for deposition of the Ag electrode and CuO film, respectively. First, p-type CuO films were deposited in an O_2_-reactive sputtering condition using a Cu target on the top epitaxial layer. After opening the chamber to change the Cu targets to Ag targets, Ag electrodes were deposited in an argon atmosphere. The sputtering conditions are presented in Table 1.

The Hall effect measurement system (HMS-5300, ECOPIA) was used to analyze the electrical properties of the CuO films. The optical properties of the fabricated films were evaluated using a UV–visible (UV–vis) spectrometer (Lambda 750 UV–vis–near infrared, Perkin Elmer). Micro-X-ray/UV photoelectron spectroscopy (UPS; AXIS-NOVA and Ultra DLD, Shimadzu) was used to investigate the work function and electroaffinity of the CuO film. X-ray diffraction (XRD; SmartLab, Rigaku) was used to evaluate the crystallographic properties. Scanning electron microscopy (SEM; S-4700, Hitachi) and atomic force microscopy (AFM; Park NX10, Park Systems) were used to investigate the surface morphology and roughness of the films at the Smart Materials Research Center for IoT, Gachon University. The thickness of the films was measured using a KLA-Tencor Alpha-Step D-500 stylus profiler.

### 2.3. p-CuO/n-β-Ga_2_O_3_-Based Photodetector with Different Postannealing Temperature

To fabricate the solar-blind photodetector, Ag/CuO layers were deposited in the shape of a circle on the upper Si-doped β-Ga_2_O_3_ epitaxial layer using a shadow mask with a radius of 300 μm. Figure 1 shows an illustration of the as-fabricated photodetector postannealed at temperatures ranging from 100 °C to 400 °C in RTA. All devices were postannealed in an argon atmosphere; the chamber was purged with 0.5/min argon gas. To evaluate the electrical and optoelectronic properties of the as-fabricated samples, a power generator (2611B, Source meter, Keithley) was used to analyze the electrical current density–voltage (*J*–*V*) and current–voltage (*I*–*V*) characteristics. The capacitance–voltage (*C*–*V*) characteristics were measured using an LCR meter (4284A LCR meter, Agilent Tech, Colorado Springs, CO, USA). A 254 nm and 365 nm UV lamp (TN-4LC, Korea Ace Scientific, Seoul, Republic of Korea) were used as the light sources to detect photoreactions from devices at different light intensities ranging from 100–1000 μW/cm^2^ at room temperature. The time-dependent photocharacteristics were evaluated under excitation at 254 nm and 365 nm at room temperature, and the transient values were recorded at zero bias.

## 3. Results and Discussion

To identify the effect of postannealing temperature on the CuO and Ag films, the structural, optical, and electrical characteristics of the Ag electrode and CuO film deposited on the glass substrate were studied before constructing the CuO/β-Ga_2_O_3_-based heterojunction photodetector. The interface conditions of the Ag/CuO layer with postannealing temperature were investigated, and the band structure and value of the built-in potential of the CuO/β-Ga_2_O heterojunction were determined using optical bandgap energy and UPS measurements. Figure 2A–D present the crystallographic characteristics and surface morphology of the CuO film as a function of the postannealing temperature. Figure 2A shows the XRD patterns of the as-fabricated CuO film and films postannealed from 100–400 °C in RTA. The as-fabricated CuO film has an amorphous structure; the postannealed samples appear at 2θ = 35.4°, corresponding to the (-111) plane (ICCD card 01-089-2531). The diffraction peaks of the CuO film exhibit sharp reflections with increasing postannealing temperature, indicating that the CuO films are more crystallized with postannealing. The Scherrer equation was used to calculate the crystallite size of the CuO films [27,28].
(1)$$\tau =\frac{K\lambda }{\beta cos\theta },$$
where *τ* is the crystallite size, *β* is the full width at half maximum (FWHM) of the peak in XRD pattern, *θ* is the Bragg angle, *λ* is the x-ray wavelength (0.15406 nm), and *K* is the Scherrer constant (0.9). The crystallite sizes of the CuO films postannealed at 200, 300, and 400 °C are 20.6, 23.2, and 24.5 nm, respectively, indicating that the crystallite size of the CuO film in the (-111) plane increases with the postannealing temperature. Figure 2B–F show the surface morphology of CuO films with different postannealing temperatures obtained from AFM with a scanning area of 1 × 1 μm^2^. The samples have a homogeneous surface morphology independent of the postannealing temperature. The root-mean-square (RMS) surface roughness was increased from 0.44 nm to 0.88 nm with increasing postannealing temperature. When the annealing temperature increased excessively, interstitial Cu atoms migrated to the surface and created oxygen vacancies that contributed to the surface roughness of the CuO film.

SEM plan-view images of CuO films with different postannealing temperatures are shown in Appendix A. All samples had a homogeneous and dense surface morphology, and no significant changes were observed in the surface of the CuO film with changes in the annealing temperature. The optical properties were measured using a UV–vis spectrometer for a specific band structure of the CuO/β-Ga_2_O_3_ heterojunction. These results are important for understanding the changes in the β-Ga_2_O_3_ band structure with respect to the postannealing temperature. The transmittance and Tauc plot of the CuO films are shown in Appendix A. There were no significant changes in the postannealing temperature. The optical bandgap energy of the CuO films was calculated using the Tauc plot equation under the assumption that the absorption product is expressed as
(2)$$\alpha h\nu =\alpha_{ο}{\left(h\nu -E_{g}\right)}^{n},$$
where $\alpha_{ο}$ is a material constant; hν is the photon energy (1240/λ); *E_g_* is the optical bandgap energy; and *n* is the power coefficient, which can be 1/2, 3/2, 2, or 3 depending on the type of transition (direct allowed, direct forbidden, indirect allowed, or indirect forbidden, respectively) [29,30]. The CuO film had a direct allowed transition (*n* = ½) and was used to calculate the optical bandgap energy. The obtained optical bandgap values were approximately 2.8 eV. The work function and electron affinity of the CuO film were investigated using UPS. The work function of the as-fabricated CuO film was 6.55 eV, and the electron affinity was 5.02 eV. The optical bandgap, work function, and electron affinity of the β-Ga_2_O_3_ and CuO film are presented in Appendix A.

The electrical properties of the CuO films were characterized to determine the electrical properties of CuO/β-Ga_2_O_3-_based photodetectors and their band structures. Figure 3A,B present the carrier concentration, resistivity, and mobility of the CuO films obtained from Hall measurements. The Hall coefficient of the CuO films was positive, indicating that the as-fabricated CuO films are p-type semiconductors. The carrier concentration of CuO films increased until 300 °C postannealing, with a maximum value of 1.36 × 10^20^ cm^−3^; the resistivity decreased until 300 °C, with a minimum value of 1.65 Ω∙cm. When the CuO film was postannealed at 400 °C, the carrier concentration decreased and the resistivity increased, as high-temperature annealing can change the CuO film composition. Cupric oxide has a copper vacancy; copper atoms ionically bond with oxygen, producing excess hole carriers [31]. Thus, the nonstoichiometric form of the CuO film can generate more hole carriers, particularly when the O content is greater than the Cu content. The energy generated by high temperatures, particularly over 300 °C, affects the energy binding of CuO, causing oxygen atoms to diffuse into the air, which lowers the oxygen ratio in the film [32]. As the stoichiometry is matched between the Cu and O content in the lattice, the carrier concentration decreases, increasing the resistivity of the CuO film when postannealed at 400 °C. The mobility of the CuO film exhibits a trade-off relationship with the carrier concentration, decreasing from 8.38 cm^2^/Vs to 2.64 × 10^−1^ cm^2^/Vs as the carrier concentration increased until 300 °C postannealing. At 400 °C postannealing, the mobility slightly increased because the carrier concentration decreased [33].

The structural and optical properties of the Ag electrode are shown in Appendix A. The surface morphology of the Ag film was investigated using the SEM image in Appendix A. Appendix A shows the X-ray diffraction (XRD) pattern of the Ag electrode on the glass substrate. The as-fabricated Ag electrode exhibits a polycrystalline structure; the Ag peaks appearing at 2θ = 38.5°, 44.8°, 64.9°, and 77.7° correspond to the (111), (200), (220), and (311) planes, respectively (ICDD card 01-087-0720). The intensity of the Ag peaks increased with increasing postannealing temperature, but no other peaks appeared and the phase did not change. Appendix A shows the transmittance of the as-fabricated Ag electrode on the glass substrate.

As the surface roughness of CuO films increases with postannealing temperature, the Ag electrode can be affected when deposited on the CuO film. Thus, the structural properties of the Ag/CuO layer were studied using the SEM images. Figure 4A shows the SEM cross-section images of the Ag/CuO layer deposited on the glass substrate. The thickness of the CuO film is approximately 100 nm; the thickness of the Ag film is approximately 20 nm. The surface morphologies of Ag/CuO films annealed at different temperatures are shown in Figure 4B–F. When the postannealing temperature was increased, the Ag films deposited on the CuO film exhibited an inhomogeneous surface structure. Two reasons are considered for the inhomogeneous growth of the Ag film on the CuO film as the postannealing temperature increased. First, the adhesion between CuO and the Ag film was affected by the rough surface morphology of the CuO film induced by increasing the postannealing temperature. A rough film surface has a lower bonding strength than a smooth and homogeneous film [34]. Second, the film adhesion decreased, owing to the different thermal stresses of the Ag and CuO films. The degree of thermal expansion of each film is considered to depend on the material. Thus, different magnitudes or directions of stress were applied to the Ag/CuO layer; the Ag film became inhomogeneous and exhibited low adhesion on the CuO film [35]. The instability and low adhesion of the Ag/CuO film can generate interfacial defects between the Ag electrode and CuO film. When the postannealing Ag/CuO layer exceeds 300 °C, the electrical properties of the photodetector are adversely affected.

Figure 5A–C show the electrical properties of the CuO/β-Ga_2_O_3_-based heterojunction photodetectors with different postannealing temperatures. Figure 5A shows the *J*–*V* characteristic curves of the as-fabricated photodetector and postannealed photodetector from 100 °C to 400 °C. Devices were measured in the dark from −2 V to 4 V. Regardless of postannealing and temperature, the rectification characteristics were confirmed for all devices. The off-current for the device was 2.17 × 10^−8^ A/cm^2^ (1.53 × 10^−11^ A). The on-current was 41.26 A/cm^2^ (2.92 × 10^−2^ A). The on−off ratio was high (1.90 × 10^9^) for the device postannealed at 200 °C; the *J*−*V* characteristics indicate that the leakage current decreases with increasing postannealing temperature. During the postannealing process, the oxygen atoms near the surface of the Ga_2_O_3_ wafer diffuse into the ohmic contact metal and form a metal–oxide compound, reducing the density of the interfacial trap between the Ga_2_O_3_ wafer and the Ti/Au contact–metal layer. Thus, the leakage current of the photodetector decreases with the postannealing temperature [36]. The C–V characteristics were estimated for the dielectric constant; all samples were measured in the dark from −4 V to 1 V at 1 MHz. The capacitance of the samples at zero bias and the C–V curve are shown in Appendix A. As the interface states play an important role in heterojunction-based devices, the trap density was investigated by space-charge-limited conduction (SCLC) from log *I*–log *V* curves in dark conditions, as shown in Figure 4B. When the applied bias is low (n = 1), the carrier concentration in the bulk is higher than that in the injected carrier from the electrode, and the current increases linearly with voltage, indicating ohmic conduction. As the applied voltage increases above a certain value, the current sharply increases (n > 3), demonstrating that all trap densities are filled by the injected carrier from the electrode. The applied voltage is defined as the trap-filled limit voltage (*V_TFL_*), which is related to the trap density:(3)$$V_{TFL}=\frac{8ed^{2}n_{trap}}{9\varepsilon_{0}\varepsilon }$$
where *q* is the electron charge, *d* is the film thickness, $n_{trap}$ is the trap density, $\varepsilon_{0}$ is the permittivity constant in free space, and *ε* is the relative dielectric constant of the film [37].

For an applied voltage greater than *V_TFL_*, a space charge is formed at the interface, restricting further injected carriers from the electrode; the current flows following trap-free behavior (n = 2), known as trap-free Child’s law, expressed as
(4)$$J=\frac{9\mu \varepsilon_{0}\varepsilon V^{2}}{8d^{3}}$$
where *J* is the current density, *μ* is the mobility of the film, and *V* is the applied voltage across the diode [37].

The trap density $n_{trap}$ is calculated using Equation (3). The trap density values at different postannealing temperatures are presented in Table 2. The trap density of the as-fabricated photodetector at room temperature decreased from 3.39 × 10^15^ cm^−3^ to 1.92 × 10^14^ cm^−3^ after postannealing at 300 °C, because the annealing process can reduce interfacial defects and dislocations between ohmic metal and β-Ga_2_O_3_. The trap density of the as-fabricated photodetector was significantly lower than that of the other heterojunction-based devices [38,39]. However, the trap density increased after annealing at 400 °C. The surface roughness of the CuO film increased with increasing postannealing temperature, and the adhesion between the Ag electrode and CuO film decreased, owing to the rough surface of the CuO film and different thermal stresses after high-temperature annealing.

Figure 4C shows the built-in potential (*V_bi_*), ideality factor (*n*), and on-resistance (*R*_on_). These parameters were calculated from the *J*−*V* curve of the heterojunction photodetector based on the thermionic emission model as follows [10].
(5)$$J=J_{S}\exp\{\frac{q\left(V-IR_{S}\right)}{nkT}\}$$
where *J_S_* is the saturation current density, *q* is the electron charge, *V* is the voltage across the diode, *R_s_* is the series resistance, *n* is the ideality factor that represents the deviation between the ideal diode and the actual diode with barrier inhomogeneity and a tunneling component, *k* is Boltzmann’s constant, and *T* is the temperature (K) [10]. *J_S_* is expressed as
(6)$$J_{S}=AA^{*}T^{2}\exp\left(\frac{-q\varphi_{B}}{kT}\right)$$
where *A** is Richardson’s constant (41 A cm^−2^ K^−2^ for β-Ga_2_O_3_), *A* is the contact area, *φ_B_* is the effective barrier height, *k* is Boltzmann’s constant, and *T* is the temperature (K).

The built-in potential of the as-fabricated photodetector increased from 0.88 eV to 1.55 eV with postannealing at 300 °C. A higher CuO film carrier concentration produced high built-in potential when contacting β-Ga_2_O_3_. With a higher carrier concentration, the closer the Fermi level of CuO is to the valence band, the greater is the difference in the Fermi level between CuO and β-Ga_2_O_3_ [26]. However, the built-in potential of the photodetector postannealed at 400 °C decreased slightly, owing to the decrease in carrier concentration. High-temperature postannealing leads to a decrease in the oxygen composition ratio of the CuO film, decreasing the carrier concentration in p-type CuO. The ideality factor of the as-fabricated photodetector decreased from 3.29 to 1.48 with increasing postannealing temperature up to 300 °C, because postannealing can reduce interfacial defects and dislocation between Ohmic contact metal and β-Ga_2_O_3._ The ideality factor was slightly increased at 400 °C because high-temperature postannealing affects the binding energy of the CuO film, reducing the oxygen ratio. These oxygen vacancies can be considered traps; accordingly, the built-in potential decreases, and the ideality factor increases. The value of *R*_on_ increases with the postannealing temperature, which is related to the condition of the interface between the Ag electrode and the CuO film. The low adhesion and interfacial defects were caused by the surface roughness of the CuO film caused by the excessive annealing temperature and different thermal stresses of the Ag electrode and CuO film [34,35]. These defects and low stability in the film interfaces can capture or restrain the movement of charge carriers, increasing the resistance [40]. In addition, the hump phenomenon of the postannealed photodetector from the *J*−*V* curve (particularly devices postannealed at 300 and 400 °C) is influenced by these defects between the Ag electrode and CuO film [40].

The band structure of the CuO/β-Ga_2_O_3_ heterojunction was investigated to further understand the electrical and photoresponse properties of the photodetector and to study the effect of postannealing on the built-in potential. Figure 5D shows the band diagrams of the as-fabricated CuO/β-Ga_2_O_3_ heterojunction photodetector and the photodetector postannealed at 300 °C. Based on the difference in the electron affinities of the CuO film and the β-Ga_2_O_3_ wafer, the conduction band offset (Δ*E_C_*) was calculated as 1.02 eV. The valence band offset (Δ*E_V_*) was estimated as 0.98 eV from optical bandgap energies of 2.80 and 4.80 eV. The CuO/β-Ga_2_O_3_ heterojunction exhibited a type-I band alignment. Another important factor of the band structure in the heterojunction-based self-powered photodetector is the built-in potential (*V_bi_*), which is determined by the position of the Fermi level of each film. In other words, the values of the built-in potential can be obtained from the difference between the Fermi levels of CuO and β-Ga_2_O_3_. After a contact is formed between the CuO film and β-Ga_2_O_3_ wafer, the Fermi level of each layer is aligned, generating band bending. Free electrons from β-Ga_2_O_3_ diffuse into the p-type CuO film and combine with the positively charged holes, resulting in neutralization of the net charge. As the diffused holes and electrons reach equilibrium, a space-charge region known as the depletion layer is formed. The depletion layer constructs an energy barrier known as the built-in potential. This energy barrier is important for self-powered photodetectors because photogenerated carriers are only transferred by an electric field in a self-powered photodetector. When light irradiates the photodetector, the energy of the incident light generates electron–hole pairs (EHPs) in the depletion layer at the interface of the CuO/β-Ga_2_O_3_ heterojunction. The photogenerated carriers are rapidly separated by the built-in potential; electrons transfer to β-Ga_2_O_3_ and holes to the CuO film. The photogenerated EHPs are more quickly and effectively separated with a greater energy barrier [12]. A greater built-in potential causes the photogenerated EHPs to move faster with a larger electric field, increasing the carrier transfer speed and reducing the probability of carrier recombination, which in turn increases the device response speed and efficiency [40]. After postannealing at 300 °C, the carrier concentration of the CuO film increased and the Fermi level was closer to the valence band. The difference between the Fermi levels of the CuO film and β-Ga_2_O_3_ increased, increasing the built-in potential from 0.88 to 1.55 eV when CuO and β-Ga_2_O_3_ came in contact. Specifically, the built-in potential can be increased at an appropriate postannealing temperature, and the photocharacteristics of the self-powered photodetector can be improved.

Figure 6A shows an illustration of the as-fabricated CuO/β-Ga_2_O_3_ heterojunction photodetector under solar-blind UV light at a wavelength of 254 nm. The photoresponse characteristics of the devices were measured under 254 nm solar-blind UV light at zero bias. The time-dependent photocurrent densities of the as-fabricated photodetectors and photodetectors annealed at different temperatures are shown in Figure 6B. The photocurrent density increased as the postannealing temperature increased to 300 °C; the maximum photocurrent density of 19.3 μA/cm^2^ was achieved for the device postannealed at 300 °C. The photo-to-dark current ratio was 1.07 × 10^3^ at zero bias under 254 nm solar-blind UV light (1000 μW/cm^2^). As the built-in potential of the device postannealed at 300 °C was higher than that of the other photodetectors, and the trap density was relatively low, the photogenerated EHPs were effectively separated by a larger built-in barrier increasing the photocurrent density. However, the photodetector postannealed at 400 °C exhibited a slightly lower photocurrent density. The low stability and inhomogeneity of the Ag/CuO layer at excessively high annealing temperatures increased the trap density of the devices. The carrier concentration of the film decreased when the CuO film was postannealed at 400 °C, which decreased the built-in potential when the CuO film formed a heterojunction with β-Ga_2_O_3_, In other words, the probability of recombination of photo-generated EHPs increased, owing to a relatively high trap density and low built-in potential [12]. The photocurrent rapidly flowed through all photodetectors as the solar-blind UV light was turned on; when the UV light was turned off, the photocurrent decreased to the dark current. All the devices suggest high reproducibility with constant photocurrent density. Figure 6B shows the photocurrent density for the photodetector postannealed at 300 °C at zero bias under 254 nm solar-blind UV illumination with light intensities ranging from 100–1000 μW/cm^2^. The time-dependent photocurrent densities of room- and other postannealing-temperature photodetectors are shown in Appendix A. The photocurrent density gradually increased as the light intensity increased because a higher light intensity generates more EHPs. The external quantum efficiency (EQE), responsivity, and detectivity were determined to examine the photoresponse characteristics of the devices. The responsivity and EQE represent the sensitivity of the photodetector; the detectivity represents the smallest detectable signal [41]. The temperature-dependent EQE of the photodetectors is shown in Figure 6D. The responsivity and detectivity were obtained from the photocurrent according to the light intensity and are shown in Figure 6E,F. The responsivity is expressed as
(7)$$R=\frac{J_{\mathrm{Photo}}-J_{\mathrm{Dark}}}{P},$$
where *J*_Photo_ is the photocurrent density, *J*_Dark_ is the dark-current density, and *P* is the supplied light intensity [12]. The detectivity is expressed as
(8)$$D=\frac{R}{{\left(2eJ\right)}^{1/2}}$$
where *J* is the dark-current density [12].

The EQE is expressed as
(9)$$\mathrm{EQE}\;=\frac{Rhv}{e}$$
where hv is the incident photon energy [42].

It was confirmed that the maximum EQE was 14.8% at zero bias under 254 nm solar-blind UV light with 100 μW/cm^2^ light intensity. The EQE equation shows the proportion of the output number of electrons to the input number of incident photons. The EQE increased with the same tendency as that of the built-in potentials. In self-powered devices, a larger built-in potential increases the speed of EHP movement and reduces the possibility of carrier recombination, increasing the photon–electron conversion efficiency [12]. Both the responsivity and detectivity increase as the light intensity decreases because more EHPs are generated with a greater light intensity. Self-heating is subsequently induced in the photodetector, increasing the possibility of carrier recombination [10]. The maximum responsivity and detectivity were 30.3 mA/W and 1.09 × 10^12^ Jones, respectively, for the photodetector postannealed at 300 °C, indicating good sensitivity of the CuO/β-Ga_2_O_3_ heterojunction photodetector. The responsivity and detectivity are superior to those of commercial Si and InGaAs-based photodetectors [43].

The response speed is a crucial performance parameter of photodetectors and is affected primarily by the magnitude of the built-in potential. The response speed was measured using the time-dependent photocurrent characteristics under 254 nm solar-blind UV light with 1000 μW/cm^2^ light intensity. The rise and fall times of the CuO/β-Ga_2_O_3_ heterojunction photodetector postannealed at 300 °C are shown in Figure 7A,B, respectively. Each time point was 12 ms and 14 ms, respectively, which is significantly faster than that in our previous study of heterojunction-based photodetectors [41]. The postannealing temperature-dependent photoresponse characteristics are presented in Table 3. As the magnitude of the built-in potential increases, the sensitivity of the device increases and the response time decreases, indicating that the built-in potential is a dominant factor in self-powered photodetectors. Appendix A plotted postannealing temperature-dependent photoresponse characteristics.

Figure 8A shows the time-dependent photocurrent density characteristics of the CuO/β-Ga_2_O_3_ heterojunction photodetector at zero bias under 365 nm UV light with 1000 μW/cm^2^ light intensity. The rejection ratio of 254 nm (solar-blind UV) to 365 nm (UV-A) was 1.80 × 10^3^ at zero bias with a light intensity of 1000 μW/cm^2^, as obtained from the time-dependent photocurrent density shown in Figure 8B. The responsivity and EQE of the devices with different postannealing temperatures at 254 nm and 365 nm at zero bias with a 1000 μW/cm^2^ light intensity are shown in Figure 8C and Appendix A, respectively. The sensitivity of UV-A light is significantly lower than that of 254 nm solar-blind UV light, indicating good selectivity of the CuO/β-Ga_2_O_3_ heterojunction photodetector. Figure 8D shows the stability of the photodetector decay. The photocurrent of the photodetector was measured once a month; the devices were stored for three months to evaluate their stability against decay. The photocurrent density decreased from 19.3 to 19.2 μA/cm^2^, demonstrating resistance to decay and oxygen degradation. This is because as-fabricated photodetector is composed of oxide semiconductor and noble metal, which allowed it to maintain the photocurrent. The photocurrent density and responsivity of the photodetector postannealed at 300 °C under reverse bias voltage (−3 V) are shown in Appendix A.

Table 4 presents a comparison of the photoresponse characteristics of the CuO/β-Ga_2_O_3_ heterojunction photodetector postannealed at 300 °C and other self-powered solar-blind UV photodetectors based on Ga_2_O_3_. The CuO/β-Ga_2_O_3_ heterojunction photodetector postannealed at 300 °C exhibits a higher responsivity and faster response speed than other solar-blind self-powered photodetectors based on heterojunctions. The results for the as-fabricated photodetector with different postannealing temperatures confirm that the magnitude of the built-in potential for the self-powered photodetector can be adjusted via the postannealing process, and that a large built-in potential allows EHPs to be quickly separated by a strong electric field, thereby increasing efficiency.

## 4. Conclusions

This study reveals an effective way to improve the photocharacteristics of self-powered photodetectors by controlling the built-in potential of photodetectors with a postannealing process. CuO/β-Ga_2_O_3_ heterojunction photodetectors with different postannealing temperatures were constructed using reactive sputtering with an FTS system. The as-fabricated photodetectors were annealed at different temperatures in RTA, and their photoresponse characteristics under illumination from 254 nm solar-blind UV light were investigated. The postannealing process affected the electrical properties and surface morphology of the CuO film; the CuO film annealed at 300 °C had the highest carrier concentration (1.36 × 10^20^ cm^−3^). The increased carrier concentration of CuO film after annealing brought the Fermi level closer to its valence band, increasing the magnitude of the built-in potential from 0.88 eV to 1.55 eV when contacting β-Ga_2_O_3_ as a heterojunction. The large built-in potential rapidly separated the photogenerated EHPs in the depletion layer, increasing the responsivity and response speed of the photodetector. The device postannealed at 300 °C exhibited a high photo-to-dark current ratio of 1.07 × 10^3^, high photoresponsivity of 30.3 mA/W, and high detectivity of 1.1 × 10^12^ Jones under 254 nm UV light at zero bias. The response speed was 12 ms for rise time and 14 ms for decay time, exhibiting good stability with aging under 254 nm UV light at zero bias. It had nearly the same current density after three months of storage in open air, indicating resistance to decay and water and oxygen degradation. These results demonstrate that adjustment of the built-in potential through postannealing significantly improves the sensitivity and response time of a self-powered heterojunction-based photodetector.

## Figures and Tables

**Figure 1 nanomaterials-13-00954-f001:**
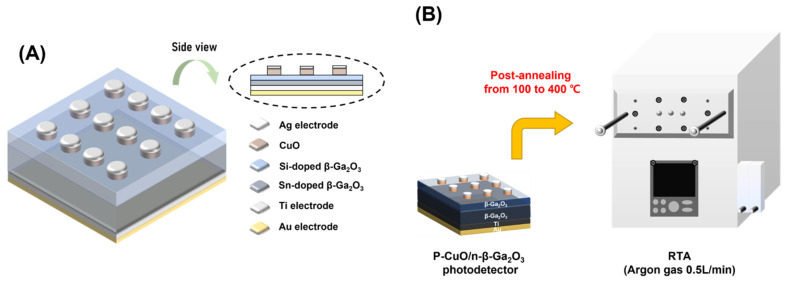
(**A**) Illustration of as-fabricated CuO/β-Ga_2_O_3_-based heterojunction photodetectors with Ag electrode; (**B**) photodetector postannealing process in RTA.

**Figure 2 nanomaterials-13-00954-f002:**
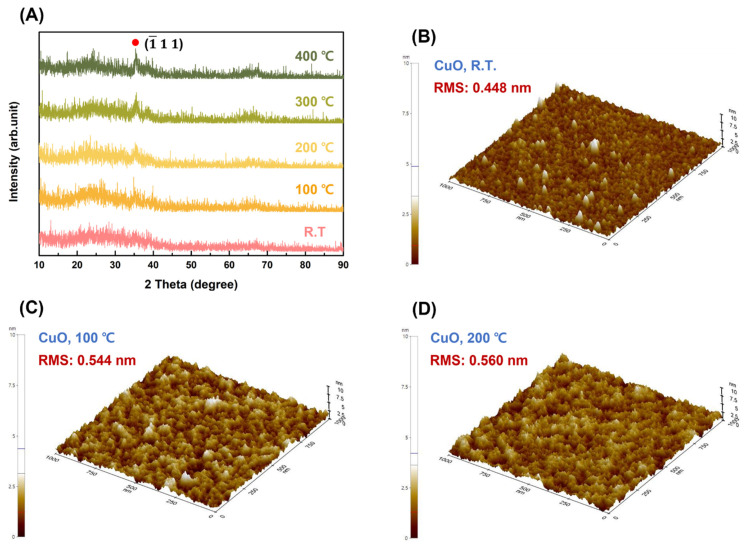
(**A**) XRD pattern of as-fabricated CuO film and films postannealed from 100–400 °C; (**B**) AFM images of surface morphology of as-fabricated CuO film and films postannealed at (**C**) 100 °C; (**D**) 200 °C; (**E**) 300 °C; (**F**) 400 °C, deposited on the glass substrate.

**Figure 3 nanomaterials-13-00954-f003:**
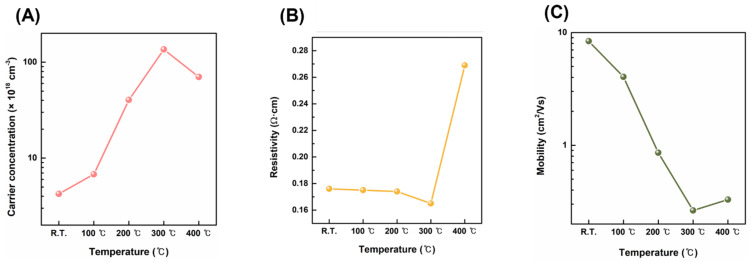
Electrical properties of CuO films with different postannealing temperatures obtained from Hall measurement: (**A**) carrier concentration; (**B**) resistivity; (**C**) mobility.

**Figure 4 nanomaterials-13-00954-f004:**
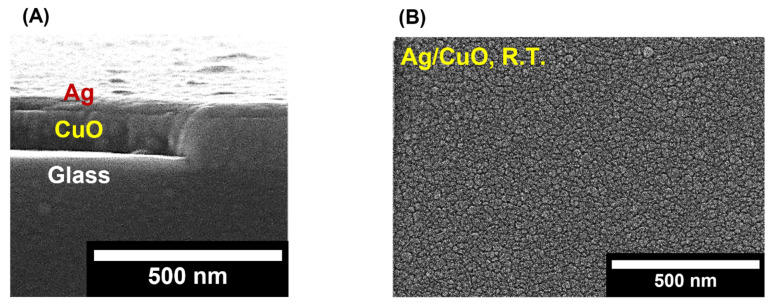
SEM images of surface morphology of Ag/CuO layer deposited on glass substrate: (**A**) cross-section image of Ag/CuO layer; plan-view images of (**B**) as-fabricated Ag/CuO layer at room temperature and postannealed at (**C**) 100 °C; (**D**) 200 °C; (**E**) 300 °C; (**F**) 400 °C.

**Figure 5 nanomaterials-13-00954-f005:**
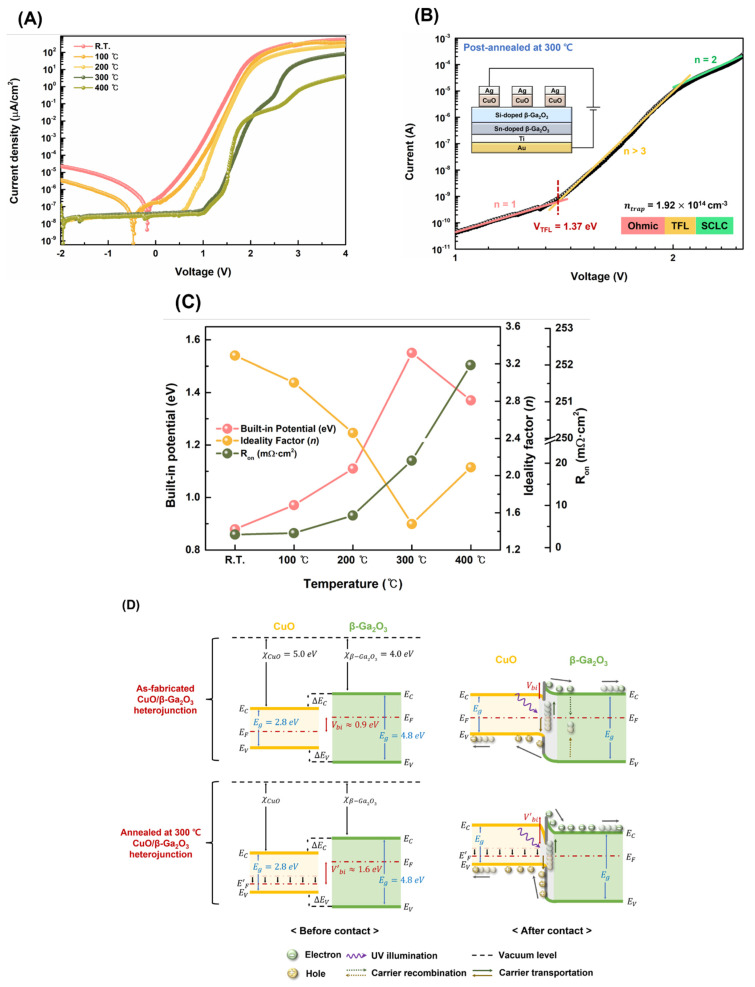
(**A**) *J*–*V* characteristic curve of CuO/β-Ga_2_O_3_ heterojunction photodetectors with different postannealing temperatures in the dark; (**B**) trap density measurements, log *I* – log *V* curve of CuO β-Ga_2_O_3_ heterojunction photodetectors with different postannealing temperatures in the dark; (**C**) variation of electrical parameters of CuO/β-Ga_2_O_3_ heterojunction photodetector with different postannealing temperatures; (**D**) experimental energy band diagrams of the as-fabricated CuO/β-Ga_2_O_3_ heterojunction photodetector and photodetector postannealed at 300 °C.

**Figure 6 nanomaterials-13-00954-f006:**
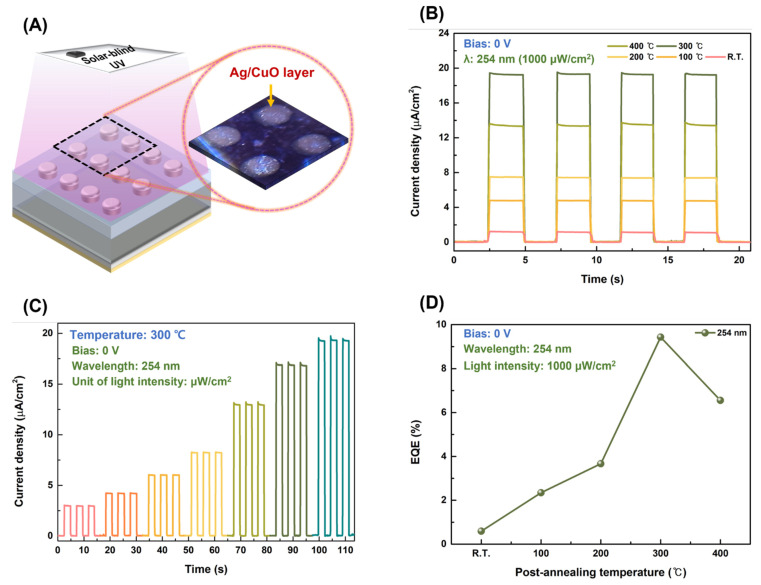
(**A**) Illustration of the as-fabricated CuO/β-Ga_2_O_3_ heterojunction photodetector under 254 nm solar-blind UV light; (**B**) continuous time-dependent photocurrent density characteristics of CuO/β-Ga_2_O_3_ heterojunction photodetectors at zero bias under 254 nm solar-blind UV light with 1000 μW/cm^2^ light intensity; (**C**) time-dependent photocurrent density characteristics of photodetector postannealed at 300 °C under 254 nm solar-blind UV light with different light intensities; (**D**) postannealing temperature-dependent EQE under 254 nm solar-blind UV light with intensity of 1000 μW/cm^2^. Variation of (**E**) responsivity and (**F**) detectivity of photodetector with varying light intensity at zero bias.

**Figure 7 nanomaterials-13-00954-f007:**
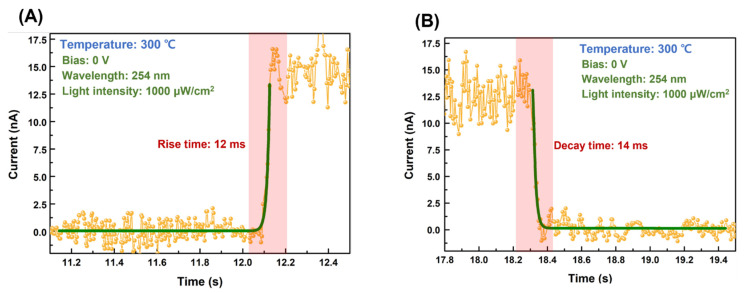
Part of time-dependent photocurrent of the CuO/β-Ga_2_O_3_ heterojunction photodetector for (**A**) rise time and (**B**) fall time at zero bias under 254 nm solar-blind UV light with 1000 μW/cm^2^ light intensity.

**Figure 8 nanomaterials-13-00954-f008:**
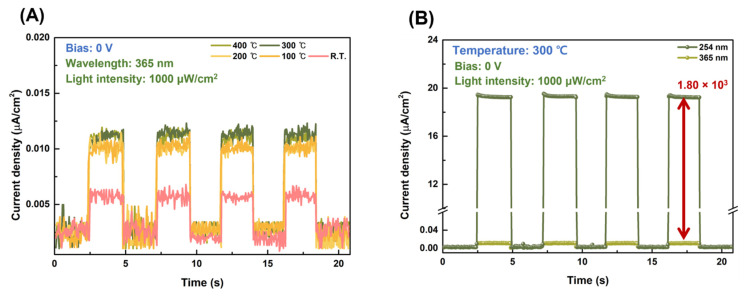
(**A**) Time-dependent photocurrent density characteristics of CuO/β-Ga_2_O_3_ heterojunction photodetectors with different postannealing temperatures at zero bias under 365 nm UV light with 1000 μW/cm^2^ light intensity; (**B**) rejection ratio of 254 nm to 365 nm light for photodetector postannealed at 300 °C; (**C**) variation of responsivity of photodetector postannealed at 300 °C with 254 nm and 365 nm light with 1000 μW/cm^2^ light intensity; (**D**) photostability test for photodetector stored for three months at zero bias under 254 nm UVC light with 1000 μW/cm^2^ light intensity.

**Table 1 nanomaterials-13-00954-t001:** Sputtering conditions for Ag electrode and CuO film.

Parameter	Conditions	
Layer	CuO	Ag
Targets	Cu (99.99%)	Ag (99.99%)
Substrate	Soda–lime glass	Soda–lime glass
Base pressure	3 × 10^−5^ Torr	3 × 10^−5^ Torr
Working pressure	2.5 mTorr	2 mTorr
Gas flow	Ar: 10 sccm, O_2_: 10 sccm	Ar: 10 sccm
Input power	150 W (DC)	15 W (DC)
Thickness	100 nm	20 nm

**Table 2 nanomaterials-13-00954-t002:** Diode parameters of CuO/β-Ga_2_O_3_ heterojunction as a function of postannealing temperature using capacitance–voltage measurements, and trap density of photodetector with different postannealing temperatures investigated via space-charge-limited conduction.

Postannealing Temperature (°C)	Capacitance (at Zero Bias) (pF)	Trap Density of Devices $\left(n_{trap}\right)$(cm^−3^)
R.T.	2.24 × 10^2^	3.39 × 10^15^
100	2.23 × 10^1^	3.65 × 10^14^
200	2.07 × 10^1^	3.42 × 10^14^
300	9.16	1.92 × 10^14^
400	1.70 × 10^1^	3.74 × 10^14^

**Table 3 nanomaterials-13-00954-t003:** Postannealing temperature-dependent values of built-in potential of devices. Photoresponse parameters of CuO/β-Ga_2_O_3_ heterojunction as a function of postannealing temperature measured at zero bias under solar-blind UV light.

Postannealing Temperature (°C)	Built-In Potential (eV)	Responsivity (mA/W) (Light Intensity: 100 μW/cm^2^)	Detectivity (×10^11^ Jones) (Light Intensity: 100 μW/cm^2^)	Rise and Decay Time (ms) (Light Intensity: 1000 μW/cm^2^)
R.T.	0.88	1.40	0.50	117/170
100	0.97	7.14	2.58	35/39
200	1.11	8.23	2.97	27/33
300	1.55	30.3	10.9	12/14
400	1.37	17.0	6.14	19/26

**Table 4 nanomaterials-13-00954-t004:** Comparison of photoresponse characteristics of solar-blind UV self-powered photodetectors under zero bias in previous studies and in this study.

Photodetector	Wavelength (nm)	Responsivity (mA/W)	Rise Time/Decay Time (ms)	Detectivity (Jones)	Reference
CuO/β-Ga_2_O_3_ (postannealed at 300 °C)	254	30.3	12/14	1.1 × 10^12^	This study
p-SiC/β-Ga_2_O_3_	254	10.4	11/19	8.8 × 10^9^	[4]
Ga:ZnO/Ga_2_O_3_	254	0.76	179/272	-	[5]
CuSCN/β-Ga_2_O_3_	254	5.50	450/260	3.8 × 10^11^	[6]
CuGaO_2_/β-Ga_2_O_3_	254	0.03	260/140	9.0 × 10^10^	[7]
NiO/β-Ga_2_O_3_	254	0.25	12/8	1.8 × 10^8^	[8]
γ-CuI/β-Ga_2_O_3_	254	2.49	-	-	[9]
Ag_2_O/β-Ga_2_O_3_	254	12.9	30/47	2.7 × 10^11^	[10]
Cu_2_O/α-Ga_2_O_3_	254	0.42	10,000/10,000	-	[11]

## Data Availability

The data that support the findings of this study are available from the corresponding author upon reasonable request.

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
