# Peer review of "A Self-Powered High-Responsivity, Fast-Response-Speed Solar-Blind Ultraviolet Photodetector Based on CuO/β-Ga2O3 Heterojunction with Built-In Potential Control"

_nanomaterials, 2023, doi:10.3390/nano13050954_

Round 1

Reviewer 1 Report

The authors have demostrated a CuO/β-Ga2O3 heterohunction photodetector by comprehensive experimental results and discussions. I have some minor questions which are listed below.

1.      Authors have analysis the crystal size of CuO films annealed at different temperatures from their XRD peaks by using Scherrer equation. But the symbols used in the equation 1 (T) and in the text () are different.

2.      The XRD patterns in Figure 2(a) seem not very good for the analysis of crystal sizes.

3.      Authors said the crystal size has been improved by annealing, but the resistivity only changed a little bit and the mobility went low seriously. I guess that this result seems not showing the interface quality has been improved by annealing.   

4.      The responsivity shown in Figure 6 indicated that the value is about 20 mA/W when the light intensity decreasing from 1000 to 200 μW/cm2. But it jumped to 30 mA/W at 100 μW/cm2. Any reasonable discussions are welcome to be addressed here.

5.      Photodetectors are usually applied by a reversed bias voltage to increase their response speed and even responsivity. In this paper authors only show the data taken at 0 bias. Is it possible to add some results taken at reversed bias voltages, even though the data in this paper are already fruitful?

Reviewer 2 Report

In this work, a self-driven solar-blind ultraviolet photodetector is designed. By using different annealing temperatures, the electronic properties of CuO film and the built-in potential of the heterojunction are adjusted, leading to the improved separation of photogenerated carriers and the detection performance of the device. Overall, the device is fully characterized and has a high responsivity and detectivity, so I suggest its acceptance after addressing the following concerns.

1.      It has been mentioned many times in the paper that the sensitivity and response time of the device can be significantly improved by changing the annealing temperature, but the detectivity, responsivity and other related parameters of the devices prepared with different annealing temperatures are not compared. It would be more convincing if the responsivity and detectivity curves of the devices prepared with different annealing temperatures could be plotted in a figure for comparison.

2.      The absorption spectra of the devices prepared with different annealing temperatures are suggested to be obtained.

3.      The introduction introduces the characteristics of gallium oxide, but copper oxide is nearly not. Can you add some reasons for choosing copper oxide as a p-type material to construct a heterojunction with gallium oxide.

4.      In the device structure, the Ag/CuO layer is deposited in a circular shape with a diameter of 300 μm. However, what is the period and what about the influence of these sizes.

5.      It is mentioned in the article that the device still maintains almost the same current density after three months of storage in an open air. Can you explain why the device shows a good stability.

Reviewer 3 Report

The paper is devoted to the development of a new, high-performance solar-blind photodiode based on the CuO/β-Ga2O3 heterojunction. The paper presents interesting results concerning the optimization of the annealing conditions for such heterostructures, aimed at improving the parameters of the photodiode. I believe that the paper can be published after clarification of some issues. There are a number of remarks:

1.    The authors report a built-in potential up to 1.55 eV. It is not entirely clear what prohibits the creation in Ga2O3 of a built-in potential comparable to the Eg of the material (4.8 eV) due to the creation of a p-n junction. Are there any restrictions on doping such materials? This issue is worth discussing in the introduction.
2.    DUV - the abbreviation is not clear. (line 103).
3.    A more detailed description of formula (1) is required: what is T (there is τ in the text)?, what is β - which curve is FWHM?
4.    It would be better to represent the dependences of the carrier concentration and mobility on the annealing temperature on a logarithmic scale. This will make it easier for the reader to grasp the scope of the changes.
5.    It is not entirely clear why the authors analyze the Ag/CuO structure grown on glass rather than as part of the complete Ag/CuO/Ga2O3 structure.

Round 2

Reviewer 3 Report

The authors duly responded to the comments. I think that the paper can be published.